# Long-Term Detection and Isolation of Severe Fever with Thrombocytopenia Syndrome (SFTS) Virus in Dog Urine

**DOI:** 10.3390/v15112228

**Published:** 2023-11-08

**Authors:** Yumiko Saga, Toshikazu Yoshida, Rieko Yoshida, Shunsuke Yazawa, Takahisa Shimada, Noriko Inasaki, Masae Itamochi, Emiko Yamazaki, Kazunori Oishi, Hideki Tani

**Affiliations:** 1Department of Virology, Toyama Institute of Health, Toyama 939-0363, Japan; yumiko.saga@pref.toyama.lg.jp (Y.S.);; 2YOSHIDA Animal Hospital, Toyama 939-0274, Japan; 3Director-General Office, Toyama Institute of Health, Toyama 939-0363, Japan; 4Department of Bacteriology, Toyama Institute of Health, Toyama 939-0363, Japan

**Keywords:** SFTSV, dog case, urine, long-term, Toyama prefecture

## Abstract

Severe fever with thrombocytopenia syndrome (SFTS) is a tick-borne infection caused by the SFTS virus (SFTSV), with a high fatality rate of approximately 30% in humans. In recent years, cases of contact infection with SFTSV via bodily fluids of infected dogs and cats have been reported. In this study, clinical and virological analyses were performed in two dogs in which SFTSV infection was confirmed for the first time in the Toyama prefecture. Both dogs recovered; however, one was severely ill and the other mildly ill. The amount of the SFTSV gene was reduced to almost similar levels in both dogs. In the dogs’ sera, the SFTSV gene was detected at a low level but fell below the detection limit approximately 2 weeks after onset. Notably, the SFTSV gene was detected at levels several thousand times higher in urine than in other specimens from both dogs. Furthermore, the gene was detected in the urine for a long period of >2 months. The clinical signs disappeared on days 1 or 6 after onset, but infectious SFTSV was detected in the urine up to 3 weeks later. Therefore, it is necessary to be careful about contact with bodily fluids, especially urine, even after symptoms have disappeared.

## 1. Introduction

Severe fever with thrombocytopenia syndrome (SFTS) is a tick-borne infectious disease first reported by Chinese researchers in 2011 [1,2], and it is caused by the Davie bandavirus (formerly, the SFTS virus; SFTSV), which belongs to the genus *Bandavirus* of the family *Phenuiviridae*. SFTS cases have been reported in China, Japan, Korea, Vietnam, and Taiwan [1,2,3,4,5,6]. Approximately 50–100 patients with SFTS are reported annually in Japan [7]. SFTS has a high fatality rate of approximately 30% in humans, making it an important public health concern in endemic areas [8,9]. As of July 2023, 900 patients with SFTS have been reported from 29 out of the 47 prefectures in Japan, gradually spreading from the western to the central parts [7]. Currently, polymerase chain reaction (PCR) tests, including quantitative PCR methods to detect the SFTSV gene, are mainly used for laboratory diagnosis of SFTS. Reverse transcription loop-mediated isothermal amplification (RT-LAMP) assays or immunochromatographic assays for antigen detection, which can provide results quickly at the bedside, are being developed; however, they are not yet practical [10,11,12,13].

Recently, it has been reported that SFTSV is transmitted to humans with tick bites and contact with the bodily fluids of SFTSV-infected companion animals, such as dogs and cats [8,14,15,16].

In the Toyama prefecture, hunting dogs have antibodies against SFTSV [17], but no human or animal cases of SFTS have been reported. In this paper, we report the first confirmed case of SFTS in two pet dogs, simultaneously detected in May 2022 in the Toyama prefecture. In this case, the diagnosis was confirmed with a PCR test using blood specimens because two out of the four dogs living together exhibited symptoms suggestive of SFTS, at the same time. In November 2022, a human SFTS case was confirmed for the first time in the Toyama prefecture [18].

To prevent the transmission of SFTSV from dogs to humans, it is necessary to obtain information such as what kind of bodily fluid SFTSV exists in and how long SFTSV is excreted from the body for. Therefore, serial clinical specimens were collected from the two dogs that tested SFTSV-positive. During the period up until the SFTSV genes and antibodies were no longer detected, the infectivity of SFTSV in each specimen was investigated. In addition, a survey of the tick habitat was conducted at the presumed infection site of the dogs, which was determined via interviewing for epidemiological information.

## 2. Materials and Methods

### 2.1. Sample Collections

Serum, urine, oral, and rectal swabs were collected from dog A (castrated male, 8 years old) from days 6 to 81 after onset (day 0). Serum and urine samples were collected from dog B (castrated male, 14 years old) from days 9 to 82 after onset. One tick collected from the body surface of dogs A and B on days 5 and 9 after onset were also used as the test material. Sera were collected from the two dogs living together with dogs A and B (spayed female, 10 years old and spayed female, 8 years old, respectively) on day 8 after the onset of disease in dog A.

In June 2022, ticks were collected from the vegetation of the low mountains in the western part of the Toyama prefecture, where the two dogs were presumed to have been infected, using the flagging method [19]. The ticks collected were classified and counted using a stereoscopic microscope.

### 2.2. Detection of SFTSV Antibody

The presence or absence of antibodies against SFTSV in the serum samples was assessed using an indirect immunofluorescence assay (IFA). The SFTSV YG-1 strain-infected VeroE6 cells were fixed in acetone. The fixed cells were then incubated with two-fold serially diluted serum samples from 40- to 80,000-fold, followed by further incubation with Goat Anti-Dog IgM or IgG H&L (FITC) (Abcam Inc., Cambridge, UK). The antibody-treated cells were visualized using a ZOE Fluorescent Cell Imager (Bio-Rad, Hercules, CA, USA).

### 2.3. Detection of SFTSV Gene

Viral RNA from serum, urine, oral, and rectal swabs collected from the dogs was extracted using a QIAamp Viral RNA Mini Kit (QIAGEN, Hilden, Germany), according to the manufacturer’s protocol. For ticks, one adult and up to five nymph ticks were pooled, and the RNA was extracted using Isogen II (NIPPON GENE, Toyama, Japan). Using the extracted RNA, the nucleoprotein (NP) gene of SFTSV was detected using real-time reverse transcription polymerase chain reaction (RT-PCR) [20], with minor modifications in three wells per sample for the dogs and in one well per sample for the ticks. The TaqMan Fast Virus 1-Step Master Mix (Applied Biosystems, Foster City, CA, USA) was used. The reaction conditions for the real-time RT-PCR were as follows: reverse transcription at 50 °C for 2 min, inactivation at 95 °C for 20 s, and, then, 45 cycles of PCR at 95 °C for 15 s and 60 °C for 1 min.

### 2.4. Virus Isolation

The clinical specimens were stored at 4 °C or −80 °C before being processed in a cell culture. The VeroE6 cells obtained from the American Type Culture Collection (Summit Pharmaceuticals International, Tokyo, Japan) were used for viral isolation and propagation. Briefly, 25 μL of each serum, rectal swab, or urine that exhibited real-time PCR positivity for SFTSV was added to the VeroE6 cells seeded the preceding day on a 24-well plate and cultured at 37 °C under 5% CO_2_. The culture supernatants were collected from the cell cultures. The viral infection dose in each supernatant was determined by the quantity of viral genomic RNA using a real-time PCR assay, as described previously [20]. The infectious viruses derived from the viral isolation method were confirmed with IFA using the Mouse Anti-SFTSV Nucleoprotein Antibody (AF10; LGC Clinical Diagnostics, Milford, MA, USA) and the Goat Anti-Mouse IgG H&L (Alexa Fluor 488) (Abcam Inc.). A viral isolation was considered negative if no viral genome was detected with the real-time PCR after three passages in the VeroE6 cells. All the viral isolation procedures were performed in a biosafety level 3 laboratory at the Toyama Institute of Health.

### 2.5. Phylogenetic Analysis of SFTSV

For genome sequencing, the RNA extracted from the specimens positive with the real-time PCR and from the SFTSV strains isolated from the urine of dogs A and B was used to amplify the NP gene of SFTSV using RT-PCR [20]. The sequencing was conducted using a BigDye Terminator v3.1 Cycle Sequencing Kit (Applied Biosystems, Foster City, CA, USA), based on the Sanger method, using a capillary DNA sequencer (3500xL Genetic Analyzer; Applied Biosystems), according to the manufacturer’s instructions. Sequence alignments were performed using the ClustalW software supported with the MEGA 6.06 software program [21]. A phylogenetic analysis was performed using the maximum likelihood method in the MEGA 6.06 software program. The statistical significance of the resulting trees was evaluated using a bootstrap test with 1000 replications. The fasta files of the sequences obtained in this study and the accession numbers of the reference sequences used in the phylogenetic analysis are given in the Appendix A.

## 3. Results

### 3.1. Epidemiological Information

The Toyama prefecture is located in the central part of Japan. SFTS cases in humans have been frequently reported in the western part of Japan (Figure 1A). Recently, the first human cases of SFTS were confirmed one after another in central and eastern Japan, including in the Aichi, Shizuoka, and Chiba prefectures (Figure 1A) [7,22]. In this study, two dogs housed indoors and outdoors (free-range in the garden), in a suburban house, in a plain area, with two other dogs living together, were found to be positive for SFTSV in the Toyama prefecture in May 2022. In addition, the dog owner owned some private land in a low mountainous area in the western part of the Toyama prefecture, and their dogs had been active in said low mountainous area (Figure 1B) 1 week and immediately before the onset of disease. The dogs had not received preventive tick treatment. Although no ticks were found inside the house, ticks were commonly observed in dogs during activities in the low mountainous area. Figure 2A shows the epidemiological information before and after the onset of symptoms in both dogs, in chronological order. One tick (*Haemaphysalis flava*, adult female) was collected from the body surface of dog A on day 5 after onset (Figure 2B), and another tick (*H. logicornis*, adult female) was collected from the body surface of dog B on day 9 after onset (Figure 2C). The SFTSV RNA was also detected in the tick samples with RT-PCR.

As these ticks exhibited mild-to-moderate blood sucking, it was presumed that they had become positive by sucking the blood of the SFTSV-positive dogs. The genetic sequence of SFTSV detected in the ticks could not be decoded; therefore, we could not confirm whether the genetic sequence was identical between the SFTSV isolated from the dogs and that isolated from the ticks. None of the caretakers or veterinary hospital staff involved with the SFTSV-positive dogs were ill. In addition, the other two dogs living in the same house appeared to be healthy, and the SFTSV gene and antibodies were not detected in either of them.

### 3.2. Clinical Symptoms and Biochemical Laboratory Findings

Table 1 shows the symptoms and laboratory findings of dog A. Dog A developed fever, vomiting, and depression on 1 May 2022 (day 0), and anorexia was observed on day 1. Subcutaneous hemorrhage, mucous and bloody stools, red urine, and protrusion of the blinker membrane in the right eye were observed on day 5. Dog A was hospitalized; however, all the symptoms disappeared on day 6. The laboratory tests revealed decreased red blood cells, white blood cells, and platelets and increased bilirubin, liver enzymes, CRP, CK, and BUN levels in the dog’s sera. On day 5, when red urine was observed, myoglobinuria due to muscle damage was suspected as the CK levels were extremely high; however, on the same day, blood in the urine was also observed. On day 5, coagulation system testing showed that the prothrombin time was within the normal range, at 7.1 s, but that the activated partial thromboplastin time (APTT) was prolonged, at 27.6 s. On day 6, the APTT returned to the normal range, at 14.3 s. Dog A was discharged from the hospital on day 15, when most of the laboratory test values were normalized.

Dog B had fever (39.7 °C), anorexia, and depression on 30 April, 1 day before the onset of disease in dog A, but his symptom disappeared in 1 day. The laboratory tests showed a decrease in the platelet counts (1.5 × 10^4^/μL) and increased CRP levels (20 mg/dL) on the onset day, but these values were normalized on day 9.

### 3.3. SFTSV Laboratory Findings

IFA was used to evaluate the changes in the antibody titers against SFTSV in the sera of the two dogs (Figure 3A). IgM and IgG were detected in the sera of two dogs, A and B, at the time of first collection (dog A, day 6; dog B, day 9), and each antibody titer peaked at this point. The IgG titers of the two dogs were almost the same; however, the IgM titer in dog A was higher than that in dog B. The copy number of the SFTSV gene in the specimens showed a similar decline in both dogs when the specimens were obtained (Figure 3B and Table 2). Low copy numbers of genes were detected in the serum and rectal swabs on days 6–12. In dog A, oral swab samples were collected on day 12; however, the viral genome was below the detection limit. In contrast, viral genomes were detected several thousand times higher in the urine of both dogs than in the other specimens. Although the copy number of the viral genome decreased 1 month after onset, the genome was detected for over 2 months or longer. In the virus isolation tests using specimens stored at 4 °C, the infectious virus was isolated in the cultured cells from the urine up to 3 weeks after the onset, when the viral genome copy was 4.6 ± 2.7 log_10_ copies/mL or higher (Figure 3C and Table 2). Infectious viruses were not isolated from the other specimens, although the viral gene was detected using PCR. However, when the freeze–thawed specimens were re-tested, the infectious virus was isolated only in the urine at day 15, with the highest genome levels, at 5.8 ± 4.9 log_10_ copies/mL. For genetic analysis, RT-PCR was performed on isolates and positive samples. The results showed that, in the RT-PCR, the specimens with a viral genome copy number higher than 5.7 ± 3.9 log_10_ copies/mL (isolates and urines up to 15 days after the onset) were positive, while the specimens with a viral genome copy number less than 4.8 ± 3.5 log_10_ copies/mL were negative (Table 2). The sequence analysis of the viruses isolated from both dogs revealed that the 453 bp sequence of the decoded NP gene was completely identical and that both belonged to the J1 clade, the most frequently reported clade detected in Japan [3] (Figure 4).

### 3.4. Tick Survey of Presumed Infected Areas

A total of forty-four ticks belonging to three genera and four species (*H. flava*, *H. longicornis*, *Amblyomma testudinarium*, *Dermacentor bellulus*) were collected from the vegetation that dogs A and B entered before the disease onset (Table 3). SFTSV was not detected using real-time PCR in the 16 RNA samples extracted from the ticks.

## 4. Discussion

In this study, we confirmed the virological diagnosis of SFTSV infections in two companion dogs and detected infectious viruses and viral genomes in their urine for a long period of time.

Of the seven dogs naturally infected with SFTSV in Japan, three (43%) had fatal outcomes [23]. However, it has been suggested that dogs can be asymptomatic or mildly ill after infection with SFTSV [23,24]. Fewer cases of SFTS have been diagnosed in dogs than in cats or humans, despite a higher rate of antibody positivity. In this study, dog B was also a very mild case, with symptoms of fever and anorexia for only 1 day, and he was diagnosed with SFTS because he was living with dog A, a severe case of SFTS. Nevertheless, the duration and amount of SFTSV shedding in dog B were similar to those in dog A, as far as the specimens which were available.

A noteworthy finding in these two cases was that the SFTSV gene was detected in the dogs’ urine at a higher concentration than in other specimens, and that the detection period for the gene was >2 months after onset. In a previous report by Park et al., SFTSV RNA was detected in the urine of dogs experimentally infected with SFTSV; however, the amount of detected RNA was similar to that found in other specimens, and its detection period was within 6 days after infection [25].

Although the cause of the difference between these cases and experimental infections is unknown, it may be due to differences in the viral strains or dogs’ age (the dogs in this case were 8 and 14 years old, and the dog used for the experimental infection was 6 months old). This is because it has been reported that SFTSV has different virulence among strains and that age is an important risk factor affecting virulence [26,27,28].

Regarding cases of SFTSV infection in dogs in the natural environment, no urine specimens were collected from other cases reported in Japan [16,23]. The SFTSV gene was detected in the urine of one out of seven dogs in the cases reported in Korea; however, the copy number was unknown because it was not quantified [29].

The laboratory findings of the viral tests of the two dogs against SFTSV suggest that urine may be a useful specimen for the detection of SFTSV in dogs. However, it is unclear whether this is a common phenomenon in dogs or whether it is just a coincidence in these two dogs. Therefore, it is necessary to answer these questions by examining more cases in dogs in the future.

In experimental infections with SFTSV in animals other than dogs, such as cats, mice, and ferrets, no clear difference was observed in the rate of detection of the SFTSV RNA levels between urine and other specimens such as blood; the detection period of the SFTSV RNA in urine was not long, <12 days after inoculation, and the difference in the detection period with other specimens was up to 6 days [28,30,31]. In addition, long-term detection of the SFTSV genome in urine has not been previously reported in human SFTS cases. However, in a human case of SFTS in which pneumonia was observed until 51 days after admission, the SFTSV RNA was detected in the sputum until 71 days after admission, whereas, in the serum, it was detected only until 16 days after admission [32]. In this case, *Candida glabrata* was also detected in the blood and *Aspergillus niger* in the alveolar lavage fluid.

In the present study, although the clinical symptoms of the two dogs disappeared on days 2 and 7 after the onset of disease, the infectious viruses were detected in the urine for at least 3 weeks after the disease’s onset. Notably, the virus isolation was unsuccessful in the frozen urine specimens, but only in the refrigerated urine specimens. This may be because freezing and thawing significantly disrupt the infectious viral particles. Based on these findings, owners and veterinarians should be careful when handling bodily fluids, especially urine, even after the symptoms have disappeared. 

One limitation of this study was that the specimens obtained were only available after symptom resolution; therefore, the dynamics of SFTSV infection in the early stages of disease onset are unknown. In human SFTS, the number of viral RNA copies in the blood decreases significantly over time [33]. Therefore, it is possible that the dogs in the present study shed more viruses in the early stages of disease onset and were at a greater risk of being a source of infection in humans.

To prevent the animal-to-human transmission of SFTSV, it is necessary to diagnose cases in companion animals at an early stage, manage them in isolation, and alert their owners as soon as possible. Therefore, it is desirable to develop and disseminate rapid diagnostic test kits that can be used in veterinary clinics, such as immunochromatography or the RT-LAMP method. It is also necessary to educate not only medical and veterinary professionals but also owners of companion animals and others about SFTS.

## 5. Conclusions

In this study, we confirmed the virological diagnosis of SFTSV infection in two companion dogs, simultaneously, and found that infectious viruses, as well as viral genomes, were detected in the urine for a long period after the symptoms had resolved. Therefore, owners and veterinarians should be careful when handling bodily fluids, especially urine, even after the symptoms have disappeared.

## Figures and Tables

**Figure 1 viruses-15-02228-f001:**
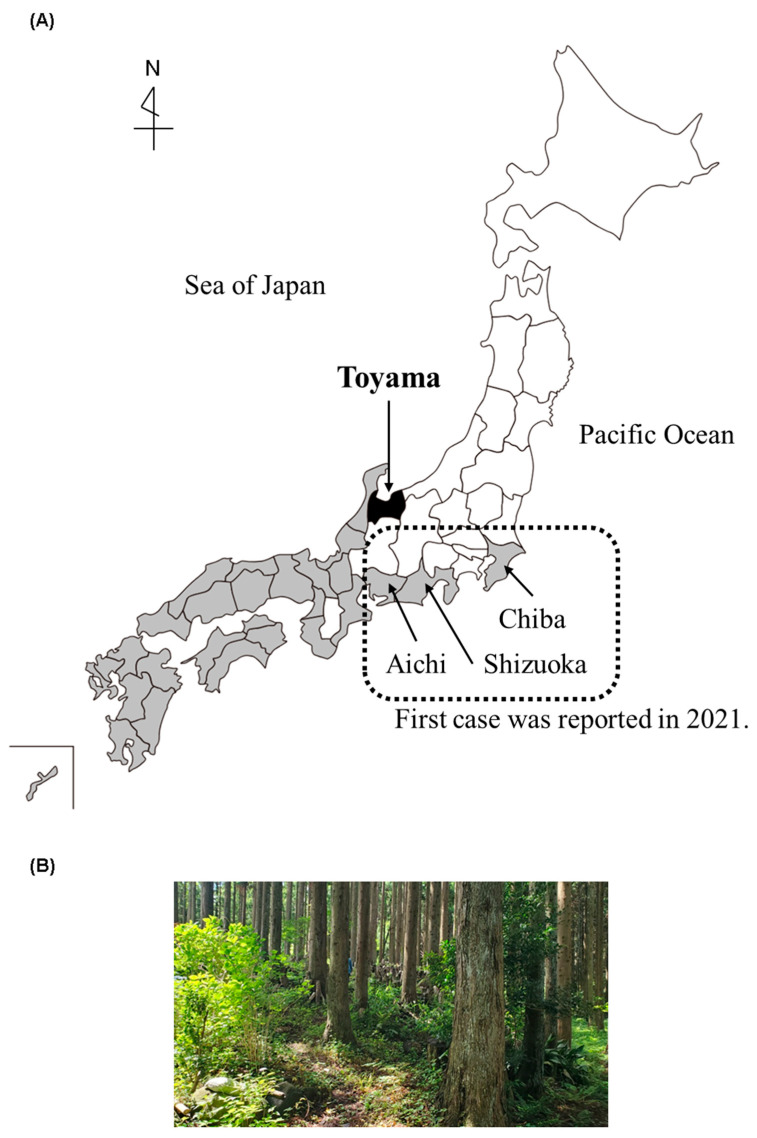
Area and environmental information. (**A**) Map of the prefectures with confirmed human SFTS cases in Japan. The black fill indicates the Toyama prefecture, where the first dog and human SFTS cases were confirmed in 2022; the gray fill indicates the prefectures with confirmed human SFTS cases in 2021, of which the three prefectures with dotted lines indicate those with the first confirmed cases in 2021. (**B**) The presumed infection site of this case.

**Figure 2 viruses-15-02228-f002:**
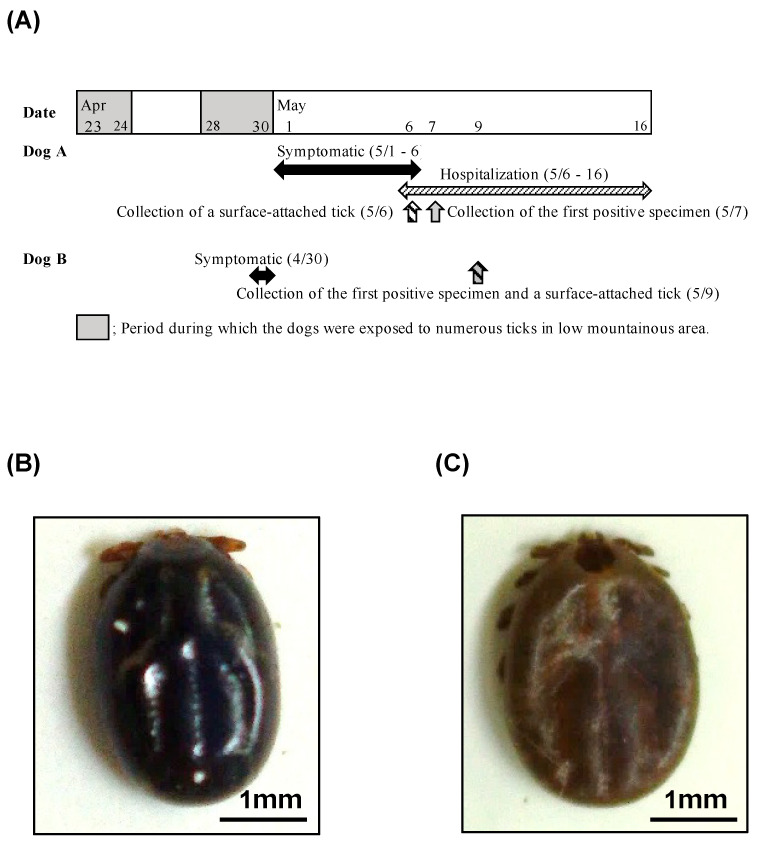
Time-series epidemiological information on this case and types of ticks. (**A**) Epidemiological information of dogs A and B in chronological order. (**B**,**C**) Ticks collected from dog A on day 5 (**B**) and from dog B on day 9 (**C**).

**Figure 3 viruses-15-02228-f003:**
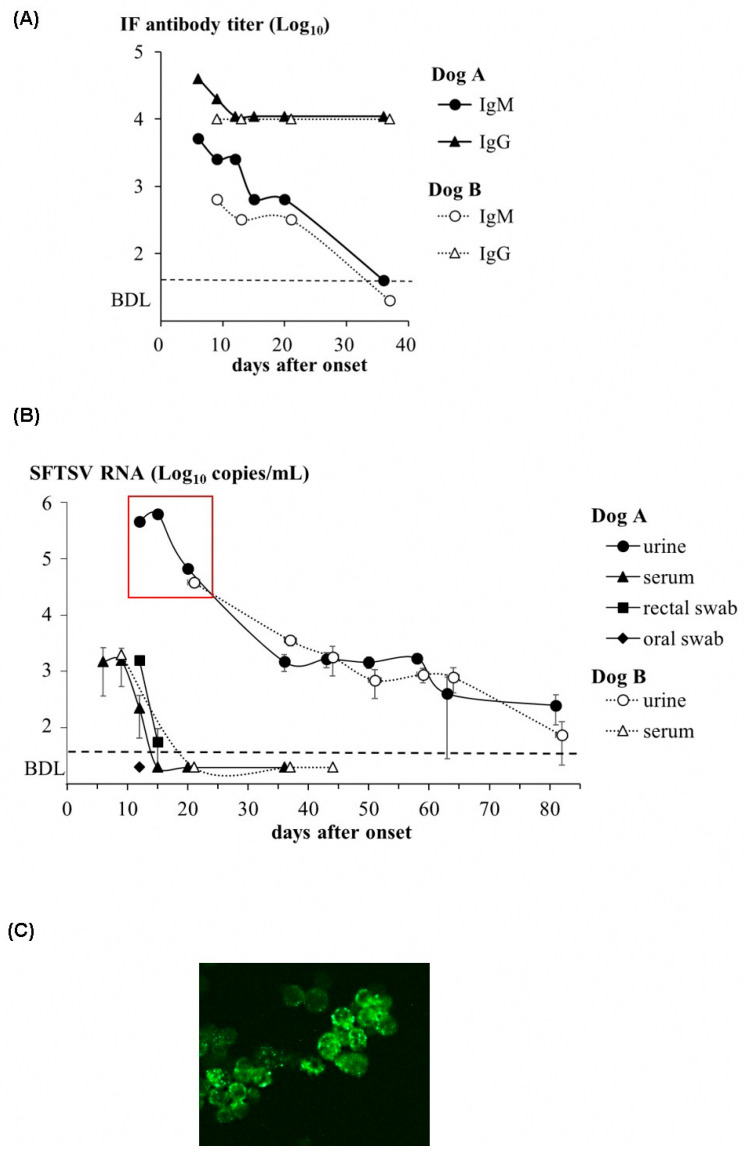
SFTSV test findings. (**A**) Antibody titers of IgM or IgG in the sera of dogs A and B with the IFA method. The detection limit is below ×40 dilution and is shown as a dashed line (BDL). The collection time is described in Table 2. (**B**) Genome copies of the SFTSV NP gene in various specimens of dogs A and B with the real-time PCR method. The detection limit is 100 copies/mL and is shown as a dashed line (BDL). The clinical specimens were collected on the indicated days after the onset which are represent in Table 2 and used for the experiment. The results shown are from three independent assays, with error bars representing the standard deviations. The specimens circled with red lines indicate those from which SFTSV was isolated. (**C**) Immunofluorescence assay for the confirmation of SFTSV isolation. The panel shows an image of the N protein of SFTSV detected in a culture of VeroE6 cells inoculated with a urine specimen collected on day 15 of dog A.

**Figure 4 viruses-15-02228-f004:**
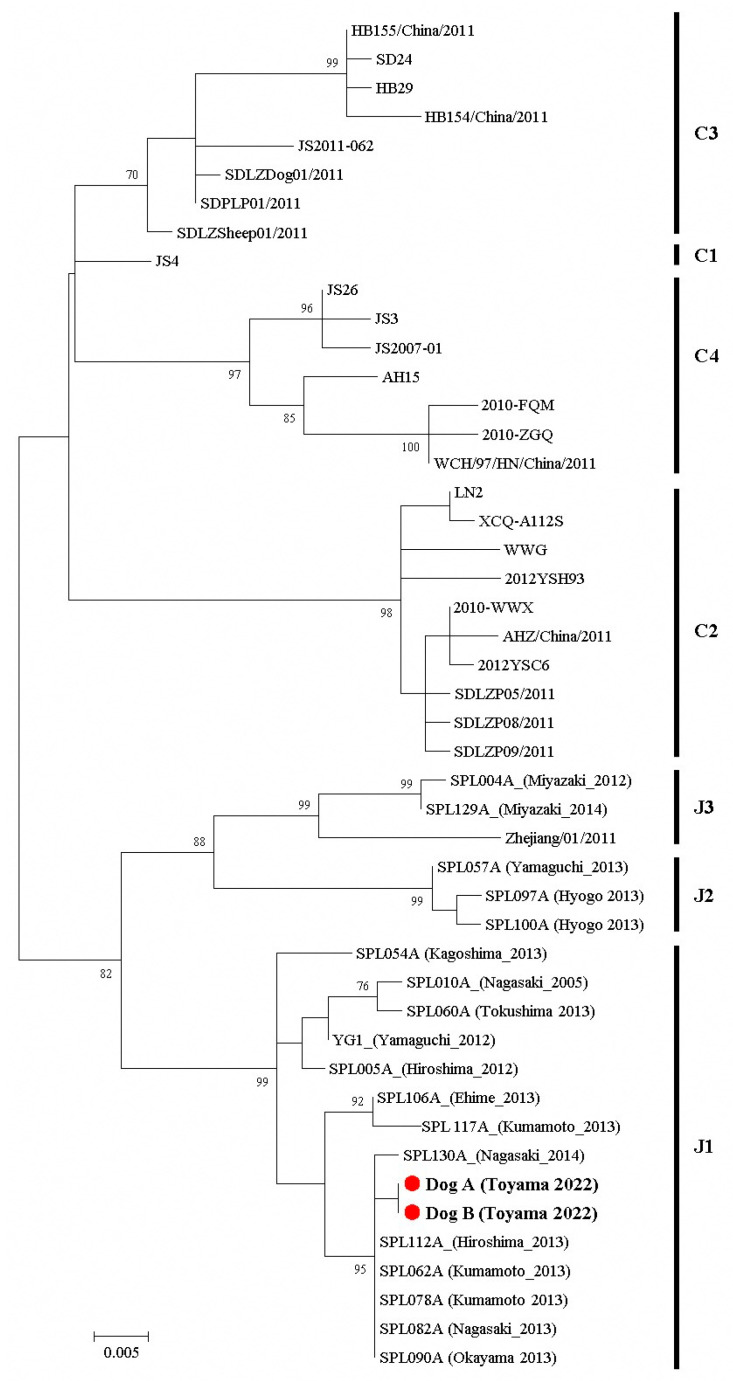
Phylogenetic analysis of the SFTSV NP gene (453 bp) of the isolated viruses from dogs A and B. The maximum likelihood trees based on the Kimura two-parameter model were constructed and tested using a bootstrap analysis with 1000 replications. The phylogenetic branchers were supported by 70% of the bootstrap values. The SFTSV strains isolated from dogs A and B were indicated with red circles and in bold. The accession numbers of the reference sequences were listed in Appendix A. J1 to J3 and C1 to C4 show the Japanese and Chinese clades, respectively.

**Table 1 viruses-15-02228-t001:** Symptoms and laboratory findings in dog A with SFTS.

Symptoms, Laboratory Findings	Normal Range	Days after Onset										
		0	1	2	3	4	5	6	8	9	12	15
Body temperature	38.0–39.0	**40.2**	**40.7**	**40.5**	**39.2**	38.0	38.1	38.8	38.0	38.0	38.2	38.6
Depression		**+**	**+**	**+**	**+**	**+**	**+**	-	-	-	-	-
Vomiting		**+**	**+**	**+**	**+**	**+**	-	-	-	-	-	-
Anorexia		-	**+**	**+**	**+**	**+**	**+**	-	-	-	-	-
Subcutaneous hemorrhage		-	-	-	-	-	**+**	-	-	-	-	-
Mucous and bloody stool		-	-	-	-	-	**+**	-	-	-	-	-
Red urine		-	-	-	-	-	**+**	-	-	-	-	-
Exposure of the blinker		-	-	-	-	-	**+**	-	-	-	-	-
RBC (10^4^/μL)	550–850	626	-	-	-	619	630	552	**517**	**466**	**394**	**429**
WBC (10^2^/μL)	60–150	**36**	-	-	-	135	135	89	**55**	**41**	126	228
PLT (10^4^/μL)	20–50	14.4	-	-	-	**2.2**	**2.7**	**5.1**	12.0	**7.1**	**2.5**	11.1
T-Bil (mg/dL)	0.1–0.4	-	-	-	-	**12.2**	**12.8**	**12.1**	**1.8**	**1.5**	**1.2**	0.9
AST (U/L)	0–52	-	-	-	-	**476**	**293**	**159**	**64**	51	49	52
ALP (U/L)	0–120	-	-	-	-	**615**	**635**	**633**	**699**	-	-	156
CRP (mg/dL)	≦1.0	-	-	-	-	-	**14**	**9.5**	**4.2**	**3.45**	**2.15**	**2.05**
CK (IU/L)	100–200	-	-	-	-	-	**>2000**	**1811**	**206**	-	-	198
BUN (mg/dL)	6–31	10	-	-	-	31	**35**	28	**33**	28	**40**	**38**
PT (s)	7.1–8.4	-	-	-	-	-	7.1	7.4	-	-	-	-
APTT (s)	13.7–25.6	-	-	-	-	-	**27.6**	14.3	-	-	-	-

RBC: red blood cell count; WBC: white blood cell count; PLT: platelet count; T-Bil: total bilirubin; AST: aspartate transaminase; ALP: alkaline phosphatase; CRP: C-reactive protein; CK: creatine kinase; BUN: blood urea nitrogen; PT: prothrombin time; APTT: activated partial thromboplastin time; +: symptomatic; -: asymptomatic or not tested; and bold: laboratory values showing abnormal values.

**Table 2 viruses-15-02228-t002:** Detection of RNA and isolation of SFTSV from the specimens of the two dogs.

Dogs	Specimens	Collection Time (Days after Onset)	SFTSV RNA (Log_10_ Copies/mL)	Results of Virus Isolation *	Conventional RT-PCR	
					N1	N2
A	Urine	12	5.7 ± 3.9	(+)	(+)	(−)
		15	5.8 ± 4.9	(+)	(+)	(−)
		20	4.8 ± 3.5	(+)	(−)	(−)
		36	3.2 ± 2.7	(−)	(−)	(−)
		43	3.2 ± 2.7	(−)	(−)	(−)
		50	3.2 ± 2.5	(−)	(−)	(−)
		58	3.2 ± 2.5	(−)	(−)	(−)
		63	2.6 ± 2.6	(−)	(−)	(−)
		81	2.4 ± 2.1	(−)	(−)	(−)
	Serum	6	3.2 ± 3.1	(−)	(−)	(−)
		9	3.2 ± 3.0	(−)	(−)	(−)
		12	2.3 ± 2.2	(−)	(−)	(−)
		15	BDL	NT	NT	NT
		20	BDL	NT	NT	NT
		36	BDL	NT	NT	NT
	Rectal swab	12	3.2 ± 2.3	(−)	(−)	(−)
		15	1.7 ± 1.6	(−)	(−)	(−)
	Oral swab	12	BDL	NT	NT	NT
B	Urine	21	4.6 ± 2.7	(+)	(−)	(−)
		37	3.6 ± 2.9	(−)	(−)	(−)
		44	3.3 ± 3.0	(−)	(−)	(−)
		51	2.8 ± 2.6	(−)	(−)	(−)
		59	2.9 ± 2.4	(−)	(−)	(−)
		64	2.9 ± 2.6	(−)	(−)	(−)
		82	1.9 ± 1.7	(−)	(−)	(−)
	Serum	9	3.3 ± 3.2	(−)	(−)	(−)
		13	BDL	NT	NT	NT
		21	BDL	NT	NT	NT
		37	BDL	NT	NT	NT

BDL: below detection limit; (−): negative; (+): positive; and NT: not tested. * The specimens used for virus isolation were stored at 4 °C and used within 8 days.

**Table 3 viruses-15-02228-t003:** SFTSV RNA detection using real-time PCR in the ticks collected from the presumed infected site of the SFTS-positive dogs.

Tick Species	Stage	Number of Individuals	Number of Positive/Tested *
*Haemaphysalis flava*	Female	4	0/4
	Nymph	14	0/3
*H. longicornis*	Female	1	0/1
	Nymph	22	0/5
*Amblyomma testudinarium*	Nymph	2	0/2
*Dermacentor bellulus*	Female	1	0/1
Total		44	0/16

* Some ticks were pooled and tested.

## Data Availability

Data used to support the findings of this study are available from the corresponding author upon request.

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
