# Peer review of "Long-Term Detection and Isolation of Severe Fever with Thrombocytopenia Syndrome (SFTS) Virus in Dog Urine"

_viruses, 2023, doi:10.3390/v15112228_

Round 1
Reviewer 1 Report (Previous Reviewer 2)
Comments and Suggestions for Authors
I am happy with the corrections (which were minor) and happy to recommend this for publication)
Reviewer 2 Report (Previous Reviewer 1)
Comments and Suggestions for Authors
I recommend to accept this manuscript.
This manuscript is a resubmission of an earlier submission. The following is a list of the peer review reports and author responses from that submission.
Round 1
Reviewer 1 Report
Comments and Suggestions for Authors
This manuscript prepared by Saga et al., reported the detection of the severe fever with thrombocytopenia syndrome virus (SFTSV) in dog urine. They found that infectious virus could be isolated approximately three weeks after the onset, and the viral genome could be detected more than two months after the onset.
Overall, the manuscript was written well, and the contents were sufficient to conclude their findings. This report will give warnings to the veterinarians who take care the SFTSV infected dogs even they recovered and do not show symptoms.
There are several points to be improved and corrected to make the manuscript easier to understand as described below.
1. Page 1, line 15, “reduced” should be better than “attenuated”.
2. Page 1, line 27 and 28, Bandavirus and Phenuiviridae should be described in Italic font.
3. Page 2, section of 2.3., the part of Detection of SFTSV gene in the Materials and Methods, please describe how many samples were run to determine and calculate the results. In Table 2, the copy number of the SFTSV genome (Log10 copies/mL) exhibited with high standard deviations.
4. Page 5, Figure 2A, the diagram could be shown in a better way. The current one is a little bit confusing. For example, put the date (month and date) on the top, and below that, put the information of the dog A and B with the date after the onset.
5. Page 8, the title of Table 2, Detection RNA should be Detection of RNA… .
Comments on the Quality of English LanguageThere is no big concern about the English. Only minor editing was asked as described in the Commnets and Suggestions section.
Reviewer 2 Report
Comments and Suggestions for Authors
A very thorough and detailed case report providing useful missing information on the etiology of this emerging zoonotic disease in domestic animals. The report is very well written and very clearly presented (which is very pleasant to see as a reviewer). The findings are straight forward and clear and the implications clearly discussed. I have no real criticisms of the paper and am very happy to recommend it for publication